# Rethinking Productivity Evaluation in Precision Forestry through Dominant Height and Site Index Measurements Using Aerial Laser Scanning LiDAR Data

Iván Raigosa-García [1], Leah C. Rathbun [2], Rachel L. Cook [1,*], Justin S. Baker [1], Mark V. Corrao [3,4] and Matthew J. Sumnall [5]

1 Department of Forestry and Environmental Resources, North Carolina State University, 2820 Faucette Dr., Raleigh, NC 27695, USA; oiraigos@ncsu.edu (I.R.-G.); jsbaker4@ncsu.edu (J.S.B.)
2 USDA Forest Service Rocky Mountain Region, 1617 Cole Blvd, Lakewood, CO 80401, USA; leah.rathbun@usda.gov
3 Department of Forest, Rangeland, and Fire Sciences, College of Natural Resources, University of Idaho, Moscow, ID 83844, USA; mcorrao@uidaho.edu or mcorrao@nmi2.com
4 Northwest Management Inc., Moscow, ID 83843, USA
5 Department of Forest Resources and Environmental Conservation, Virginia Polytechnic Institute and State University, Blacksburg, VA 24061, USA; msumnall@vt.edu
* Correspondence: rlcook@ncsu.edu; Tel.: +1-919-515-5979

**Abstract:** Optimizing forest plantation management has become imperative due to increasing forest product demand, higher fertilization and management costs, declining land availability, increased competition for land use, and the growing demands for carbon sequestration. Precision forestry refers to the ability to use data acquired with technology to support the forest management decision-making process. LiDAR can be used to assess forest metrics such as tree height, topographical position, soil surface attributes, and their combined effects on individual tree growth. LiDAR opens the door to precision silviculture applied at the tree level and can inform precise treatments such as fertilization, thinning, and herbicide application for individual trees. This study uses ALS LiDAR and other ancillary data to assess the effect of scale (i.e., stand, soil type, and microtopography) on dominant height and site index measures within loblolly pine plantations across the southeastern United States. This study shows differences in dominant height and site index across soil types, with even greater differences observed when the interactions of microtopography were considered. These results highlight how precision forestry may provide a unique opportunity for assessing soil and microtopographic information to optimize resource allocation and forest management at an individual tree scale in a scarce higher-priced fertilizer scenario.

**Keywords:** LiDAR; site index; precision forestry





## 1. Introduction

### 1.1. Loblolly's Pine Importance

Loblolly pine (*Pinus taeda* L.) is the most prevalent planted forest species in the southern US and covers 14.4 million planted hectares across 13 states [1]. In the context of plantation management, site index is a measure of site quality that reflects the potential maximum height productivity of a fully stocked, even-aged stand, and is generally considered to be independent of stand density [2].

### 1.2. Dominant Height as a Measure of Productivity

For any given tree, total height is directly related to the tree's above-ground biomass and total volume, while merchantable height is based on local specifications for given wood products [2–4]. Dominant height is defined as the average total height of the dominant and codominant trees for a given area and is often used to calculate site index [5].

Due to the difficulty in measuring the total height for all trees within a stand, different definitions for dominant height have been created, largely founded on field-plot sampling methodologies. One of the earliest definitions of dominant height was the average height of the tallest one hundred trees per hectare [6,7]. Other sources define dominant height as the average height for the largest one hundred trees per hectare, determined by the greatest diameter at breast height (DBH), measured at 4.5 feet from the ground) [8] In general, definitions for dominant height for a given area can be placed into three categories [9]:

1. Mean height for a given percentage of the tallest trees;
2. Mean height for a given number of the tallest trees as defined by size/density;
3. The tallest single tree height.

Regardless of the definition used, sampling methods are necessary to obtain an estimate of dominant height. These sampling methods can introduce bias when the tallest trees are unevenly distributed across the stand [10]. A suggested approach to correcting this bias involves the random placement of large field plots (>1 ha) to measure at least 100 trees per plot, across the stand [8]. With regard to field sampling, bias can be addressed if the plot size can appropriately capture the spatial distribution of tree heights within the given area [8,11].

*1.3. Stand Characterization with Remote Sensing Data*

Remotely sensed data such as satellite imagery or LiDAR data provide a unique opportunity to assess dominant height and site index at finer resolution scales than a stand level, further classifying a stand into multiple strata to inform the estimations of the site index. Ultimately, allowing forest managers to obtain a more detailed understanding of how tree growth varies across different micro-topographical positions within a stand will create opportunities for increased management action effectiveness that may lead to cost savings. The acquisition of large-scale, high-resolution ALS data additionally provides an opportunity to advance the role of precision forestry in assessing site quality and forest inventory; however, the standardization of a definition for "dominant height" across any region is needed given the new era of big data in forestry.

Remotely sensed data can mitigate some challenges related to sampling for dominant height within a stand. For example, ALS point clouds can provide wall-to-wall geospatial measurements and the locations of tree canopies within a stand. It is also possible to use ALS point clouds to estimate additional metrics, such as crown length and height to live crown, for individual trees, as well as total stand biomass [2,12–16]. Commonly applied methods for determining total tree height from ALS data include the identification of the highest LiDAR returns within a pre-defined area assumed to represent an individual tree crown [16]. However, total tree height estimation using individual tree crown (ITC) methodologies from ALS data can provide results that are comparable to field measurements [15,17–19]. For example, Sumnall et al. [15] found an overall R2 value of 0.87 and a Normalized Root Mean Square Error (NRMSE) of 6.52% between field measurements and LiDAR-derived measurements for total tree height. With the ability of ALS to provide wall-to-wall coverage and more accurate estimates of individual total tree height than field measurements within a stand, the definition of dominant height can be reexamined across different scales. In most situations, estimating DBH from ALS data is still a challenge as data density plays a significant role in model accuracy and tends to decrease with closer proximity to the ground [12]. To solve this point density issue, terrestrial laser scanning (TLS) has been used to estimate DBH and individual tree volume [13,20–22]; however, scaling these methods from a few hectares to full stands or the landscape scale has been challenging [20]. Nevertheless, the use of the ultra-high-density drone LiDAR can produce clouds with thousands of points per square meter, with low-altitude flights and wide scan angles that resolve individual stem and branch structures [20].

### 1.4. Use of Remote Sensing in Precision Forestry

Precision forestry uses information collected from various sensors to optimize spatial and temporal inputs aimed at maximizing forestland productivity. The goal of precision forestry is to improve stand productivity and operational efficiencies through high-resolution data [23]. Several data sources can be used in precision forestry, including satellite imagery, photogrammetric detection and ranging (PhoDAR), and light detection and ranging (LiDAR) data. Airborne laser scanning (ALS), which is one method of collecting LiDAR data, penetrates the canopy from above and provides feedback about the bare earth, forest structure, and understory cover. LiDAR data can be used to measure individual tree height, which can then be integrated with existing biometric equations [24] to estimate stand- or tree-level metrics, ecosystem services such as $CO_2$ sequestration, and other attributes that are important for economic valuation and management planning. Moreover, when combined with precise digital terrain models (DTMs) and soil maps, these high-resolution data products provide a unique opportunity to investigate the drivers of tree height at multiple scales and could replace costly ground-based forest inventory measurements. Furthermore, ALS produces high-quality metrics that can be used for creating and validating allometric models that inform the decision process of forest management [17,18].

The goal of this study is to enhance the characterization of forest stand productivity by assessing forest inventory methodologies that can benefit from the use of LiDAR-derived individual tree heights and better within-stand variability measurements to inform stand-level dominant height and site index estimates for loblolly pine plantations located across the southern USA. Our objectives were as follows:

1. Assess how scale impacts these estimates for loblolly pine plantations;
2. Compare different definitions of dominant height;
3. Evaluate methods for stand stratification (e.g., soils, topography, and soils plus topography), relative to simulations of traditional field inventory sample plot data.

This study makes several contributions to the literature. First, we provide an application that uses ALS LiDAR and topographic data to characterize within-stand productivity for loblolly pine plantations—a globally important supply source of timber. Second, our analysis demonstrates that while dominant height definitions may be consistent, the percentile definition does a better job representing dominant height in contrasting tree densities. The study also reveals that soil and topography play an important role in dominant height, even at the sub-stand level, and should not be overlooked. We also develop and demonstrate a method to stratify and weight site productivity, and provide tools to incorporate soils and microtopography in management decisions. Finally, we discuss the implications of our findings for forest managers interested in precision silviculture.

## 2. Materials and Methods

### 2.1. Site Locations

Three study sites were located in the southeastern United States, inclusive of Texas, Alabama, and North Carolina (Figure 1). Eight separate forest stands were captured within these study sites. The sites in Texas and Alabama each contain two stands, with four additional stands located in North Carolina. Latitude ranges from 30°32″ N to 36°15″ N and longitude from 94°43″ W to 77°58″ W. Stands at each site are close in proximity, under 10 km apart.

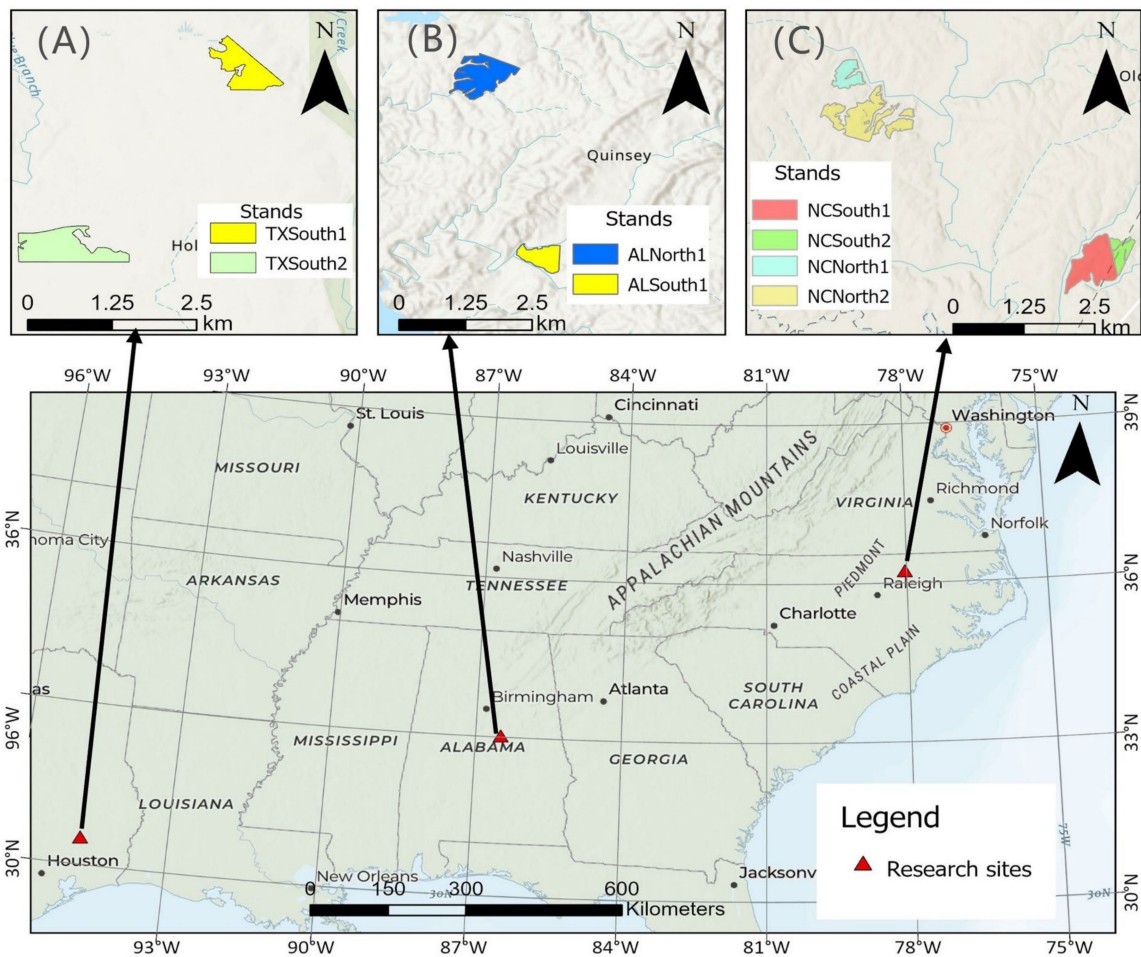

**Figure 1.** Location of the three study sites. Study site locations are indicated with a red triangle and stand locations are identified in the insets ((**A**)—Texas; (**B**)—Alabama; (**C**)—North Carolina).

Across this region, mean annual precipitation ranges from 1298 to 1154 mm and the mean annual temperature from 20.1 °C to 15.6 °C (Table 1). The mean annual maximum temperature varies from 35.6 to 33.1 °C, and the mean annual minimum temperature from 3.4 °C to −2.1 °C [25].

**Table 1.** Climate data for each study site covering the years 1981 to 2022 [25].

| Location | Mean Annual Precipitation (mm) | Mean Annual Temperature (°C) | Mean Annual Maximum Temperature (°C) | Mean Annual Minimum Temperature (°C) |
|---|---|---|---|---|
| Alabama | 1298 | 17.3 | 33.8 | 0.2 |
| North Carolina | 1154 | 15.6 | 33.1 | −2.1 |
| Texas | 1241 | 20.1 | 35.6 | 3.4 |

All stands were planted with loblolly pine between the years 1994 and 2005 and varied in size from 13 to 56.5 hectares (Table 2). The majority of stands received a thinning at 16 to 19 years after establishment; two stands, one in Texas and one in North Carolina, received a thinning at 22 and 27 years after establishment, respectively. Across all stands, thinning treatments included removing every fifth row. Residual density ranged between 298 and 733 trees per hectare. The stand-level difference in elevation varies between 7.2 and 41.4 m, with the stands in Alabama located at higher elevations than those in Texas or North Carolina.

**Table 2.** Stand overview information including management actions and elevation.

| State | Stand | Area (ha) | Year Planted | Age at Treatment (Years) | Tree Density after Thinning (Trees/ha) | Elevation Range above Sea Level (m) |
|---|---|---|---|---|---|---|
| AL | ALNorth1 | 49.1 | 2001 | 19 | 733 | 127.1–168.5 |
| | ALSouth1 | 20.4 | 2002 | 18 | 941 | 118.8–146.0 |
| NC | NCNorth1 | 16.7 | 2003 | 18 | 443 | 58.8–88.9 |
| | NCNorth2 | 56.5 | 2005 | 16 | 403 | 55.7–88.9 |
| | NCSouth1 | 51.5 | 1994 | 27 | 298 | 52.0–78.9 |
| | NCSouth2 | 13.0 | 2004 | 17 | 401 | 60.4–79.5 |
| TX | TXSouth1 | 52.3 | 2004 | 17 | 577 | 49.3–57.4 |
| | TXSouth2 | 35.8 | 1999 | 22 | 411 | 42.0–49.2 |

*2.2. Soils and Geology*

Soil maps were provided by the Forest Productivity Cooperative (FPC) Site Productivity Optimization for Trees (SPOT) system's soil code [26]. These soil maps provide a comprehensive assessment of factors known to affect tree response to forest management practices because the SPOT codes characterize soil based on a combination of factors such as dominant profile texture, depth to an increase in clay, drainage class, three soil modifiers, geology, and physiographic province. Data sources include the SSURGO database, which is an effort deriving from more than 100 years of work [27]. The sites located in Alabama are in the southern Piedmont and contain igneous, sedimentary, and metamorphic rocks that contain calcium (Ca), iron (Fe), potassium (K), and magnesium (Mg). They have a fine or coarse loamy texture with varying degrees of drainage ranging from somehow poorly to well drainage. The North Carolina sites are situated in the transitional area between the Piedmont and the Atlantic coastal plain. Soil texture varies from silty to fine loamy and is well-drained. The sites located in Texas are in the Gulf coastal plain physiographic province and the Lissie formation; these soil units were found to be severely deficient in phosphorus and have a coarse or fine loamy texture that varies from poorly drained to moderately well drained (Table 3).

**Table 3.** Soil occurrences described by the FPC SPOT codes at each stand location.

| Site | Stand | Soil Type (FPC SPOT Codes) |
|---|---|---|
| AL | ALNorth1 | B4SyxfSrPD and C2WextScPD |
| | ALSouth1 | B1WeaoBgPD, B4SyxfBgPD, C2WextBgPD, and C2WextScPD |
| NC | NCNorth1 | B2WekoAuPD and E2WykoAuPD |
| | NCNorth2 | B2WekoAuPD |
| | NCSouth1 | B2WexvAuPD, B3WekoAuPD, E2WykoAmPD, and E2WyxvAuPD |
| | NCSouth2 | B2WexvAuPD, B3WekoAuPD, and E2WyxvAuPD |
| TX | TXSouth1 | B3WeioLbGF, C3MeioLbGF, C3PyaoLbGF, and C3WeioLbGF |
| | TXSouth2 | C2MeioLbGF, C3MeioLbGF, C3PyaoLbGF, C4WoioLbGF, C6MyioLbGF, and C6PyanLbGF |

*2.3. Data Collection and Processing*

Data were collected using multiple aerial LiDAR sensors across the three study sites over the years 2020 and 2021. Table 4 provides the details for the LiDAR acquisitions. The choice of sensors was dictated by data availability rather than a deliberate intent to compare various sensors simultaneously. All the flights were conducted in a parallel pattern at 400 m above the ground in Alabama, 180 m above the ground in Texas, and 60 m above the ground in North Carolina. All airborne acquisitions used onboard aircraft navigation and real-time kinematic geographic positioning system with a precision of

≤10 cm and positional accuracy of ≤5 cm using RiACQUIRE [28]. Re-processing tasks, such as calibration scan data and strip adjustment, were performed in RiProcess [29].

**Table 4.** LiDAR acquisition details including dates of collection, and sensor and flight information. For more details and outcomes from the flight in North Carolina, please see Sumnall et al. [15].

| Location | Dates of Collection | Sensor | Pulse Density | Flight Line Overlap |
|----------|---------------------|--------|---------------|---------------------|
| AL | January 2020 | Riegl 1560ii | 26 pulses/m$^2$ | 60% |
| NC | July 2021 | Riegl MiniVux1 | 20 pulses/m$^2$ | 80% |
| TX | January–March 2021 | Riegl VQ-480 II | 289 to 493 pulses/m$^2$ | 55% |

All analyses were completed within R v.4.1.2 [30]. The LiDAR point cloud was processed to ensure all outliers were removed using the LidR package [31]. We applied the statistical outlier removal (SOR) tool to identify outliers in the point cloud (e.g., extreme ground points and extreme canopy points). Filter parameters for this process included the use of one normal distribution and a k-nearest neighbor value of eight points [31]. The LAScatalog processing engine was used to segment the LiDAR point cloud into smaller data files [32] and default parameters for segmentation size were used. After outliers were removed, we processed the LiDAR point cloud to create two products used in the analysis: (1) the digital terrain model (DTM) and (2) the individual tree crown (ITC) segmentation layer.

We performed ground classification of the LiDAR point cloud using the classify ground function in R, with a progressive morphological filter with window sizes of 3, 12, and 3 m and elevation difference threshold sizes of 0.1 and 1.5 m [32,33]. The resulting DTM represents a pixel size of 0.25 m. We generated the DTMs using the grid_terrain function and the TIN algorithm within the LidR package. Using the DTM, we calculated the difference between elevation at a central point and the average elevation around it for a given radius or topographical position index (TPI) [34] for each 0.25 pixel. To calculate TPI, we used the TPI function within the spatialEco package [35] with an adaptive window size. The size of the window was different for each site and based on the rate of elevation change for a given unit area. Hence, we used a smaller window size for sites with a smaller elevation change and a larger window size for sites with a greater elevation change. We performed a sensitivity analysis to determine the correct window size by conducting comparisons between the TPI raster file and known topographical structures within the stands. The sensitivity analysis indicated that an ideal window size was correlated with ten times the range of elevation within the stand.

From the DTM, a topographic position index (TPI) layer was created to describe three microtopography classes: valley, flat, or ridge. For each pixel in the DTM, microtopography classes were calculated [34] as follows:

1. Valleys—value of less than one negative standard deviation of the TPI;
2. Flat—value greater than or equal to one negative standard deviation but less than one half standard deviation of the TPI;
3. Ridge—value greater than one half standard deviation of the TP.

We delineated individual tree crowns (ITC) within the LiDAR point cloud using the ITC segmentation method created by Sumnall et al. [15] for loblolly pine. The detection of stem locations in this method relies on exploiting the high pulse density of the LiDAR data. ITCS delineation is based on segmenting the canopy height model (CHM) [12] using previously detected stems as initial markers. The author reported an average proportion of missed alive trees of around 8%; however, for low tree densities (618 trees/ha), the method only missed 1–3%. For each tree crown, we used the tallest returned pulse to determine the total height and the center of the tree. A microtopography class was assigned to each tree center. Microtopographic classifications were determined as the mode value for all pixels

contained within a 2.76 m radius circle around the tree center using the majority option in the exact_extract function from the exactextractr package [36]. In addition, for each tree center, we assigned a corresponding FPC SPOT soil code. We removed all trees contained within a buffer zone of 15 m from the edge of each stand to address any edge effects on tree height growth for this analysis.

*2.4. Data Analysis*

2.4.1. Dominant Height Definitions

Three definitions of dominant height were calculated for each stand using all trees identified from the ITC methodology. In addition, plots were simulated for each stand and the dominant height values were determined for trees found within each plot. The simulation included the generation of a grid of 150 m in size randomly placed within each stand, with a 12.6 m radius circular plot (1/20 hectare) at the "node or centroid" of each grid cell (total plots within each stand are provided in Table 5). The size and shape used in the simulation are common in forest inventories for pine plantations [37]. Bar graphs were created to assess the differences across the dominant height definitions for each stand and the plot simulation. Based on these differences, one definition was selected for further analysis of how scale impacts dominant height and site index. The definitions for dominant height were as follows:

1.  Top HT 85—mean height of the 85th percentile of tallest trees;
2.  Top HT 100—mean height of the tallest 100 trees per hectare within the stand;
3.  Top HT 40—mean height of the tallest 40 trees per hectare within the stand.

**Table 5.** Summary of LiDAR-derived population tree detection value results by stand, including the size of the stand, total and mean trees detected, and mean tree height with the corresponding standard deviation.

| Location | Stand | Area (ha) | Number of Trees | Mean Trees per Hectare | Mean Tree Height (m) | Number of Plots Simulated |
|---|---|---|---|---|---|---|
| AL | ALNorth1 | 49.1 | 35,537 | 724 | 17.3 (1.9) | 22 |
| | ALSouth1 | 20.4 | 19,154 | 938 | 17.2 (1.6) | 9 |
| NC | NCSouth1 | 51.5 | 15,136 | 294 | 21.2 (1.7) | 26 |
| | NCSouth2 | 13.0 | 5260 | 404 | 17.0 (1.4) | 9 |
| | NCNorth1 | 16.7 | 7296 | 437 | 17.3 (1.5) | 9 |
| | NCNorth2 | 56.5 | 22,723 | 400 | 17.2 (1.7) | 31 |
| TX | TXSouth1 | 52.3 | 19,061 | 365 | 17.7 (1.2) | 24 |
| | TXSouth2 | 35.8 | 18,585 | 519 | 22.0 (2.0) | 16 |

2.4.2. Scale

To assess the effects of scale on dominant height, each stand was stratified into four scales:

1.  Full stand: dominant trees are selected from the tree population for each stand without stratification.
2.  Soil classification: dominant trees are selected from the tree population for each soil type.
3.  Microtopography classification: dominant trees are selected from the tree population for each microtopography type.
4.  Combined soil and microtopography classification: dominant trees are selected from the tree population for each combination of soil and microtopography.

Each tree center found from the ITC methodology was classified for each scale and the area within each stand was determined for each classification (Figure 2).

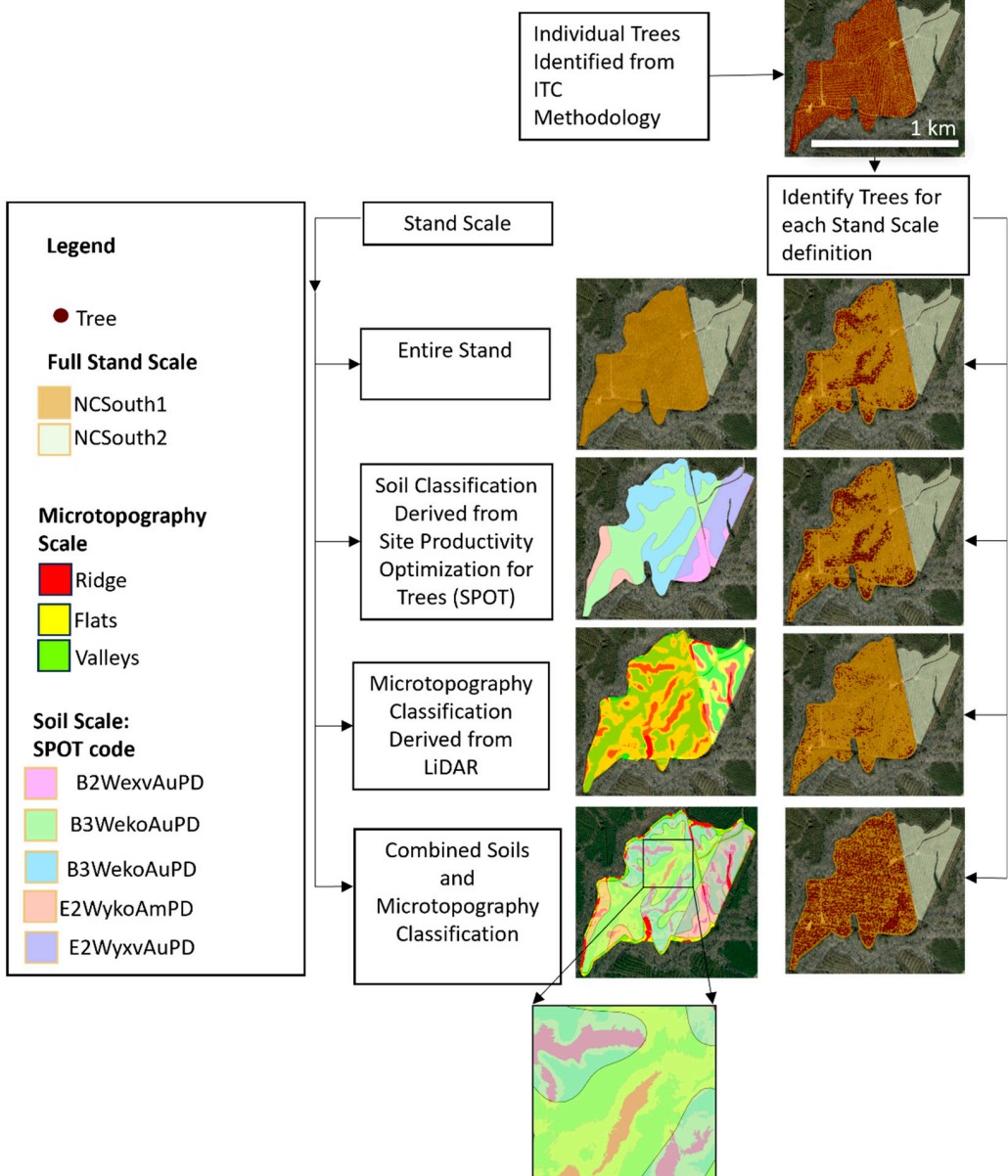

**Figure 2.** Stand scale and tree selection process for stand NCSouth1.

To explore the impacts of scale on dominant height, dominant height was calculated for each class within each scale (e.g., soils, microtopography, and the interaction of soils and microtopography). Boxplots were created to compare the range of tree heights used to compute the dominant height value for each soil type for each stand. These boxplots were compared to the trees used at the full stand scale. In addition, Dunnett's tests [38] were conducted to determine if there was a difference between the means for each soil class compared to the full stand values for each stand. The tests were conducted at the stand level to ensure the effect of age was removed. Dunnett's tests were also conducted to compare each microtopography scale to the full stand values for each stand. For each test, an alpha level of 0.05 was used to determine statistical significance.

To investigate the interaction of soils and microtopography, we developed heat maps for each study site. The heat maps visualize the percent differences between the mean dominant height values at the combined soil and microtopography scales compared to the full stand scale. Percent difference was used to normalize the information across the [39] different aged stands.

2.4.3. Site Index

To assess the impacts of incorporating scale into stand-level site index estimates, a weighted site index was calculated for each scale and compared to the full stand estimate for the site index. The site index was calculated using the equation for loblolly pine by Dieguez-Aranda et al. [39]:

$$SI = \frac{26.14 + X_0}{1 + \frac{1455}{X_0 t^{-1.107}}} \tag{1}$$

where $t$ is a base age value of 25 and $X_0$ is as follows:

$$X_0 = \frac{Y_0 - 26.14 + \sqrt{(Y_0 - 26.14)^2 + 4 * 1455 Y_0 t_0^{-1.107}}}{2} \tag{2}$$

where $t_0$ is the stand age and $Y_0$ is the dominant height. Site index values were computed for each class within a scale and a weighted average was determined for each stand. Weights were based on the area for each class within a scale:

$$\text{Weighted } SI = \sum_{i=1}^{n} \frac{SI_i A_i}{\sum\limits_{i=1}^{n} A_i} \tag{3}$$

The Wilcoxon test [40] was conducted to determine if there was a difference among weighted site index estimations across all stands for each scale. For each test, an alpha level of 0.05 was used to determine statistical significance.

## 3. Results

The number of population trees detected from the LiDAR point cloud varied from 294 to 938 per hectare, with North Carolina having the lowest-density stands and Alabama having the highest (Table 5). Mean tree height was consistent at around 17 m for the majority of the stands, except for two, NCSouth1 and TXSouth2, which were 21.2 and 22.0, respectively. These two stands are also the oldest in the study. The standard deviation of mean tree height varied from 1.2 m to 2.0 m across all stands, while the total number of trees varied from 5260 to 35,527 per stand.

### 3.1. Dominant Height Definition

The dominant height definitions impact the dominant height value and its corresponding variability (Figure 3). As expected, when dominant height is defined by a given number of trees per unit area, the mean value increases when the given number of trees per unit area decreases. This trend is seen across all stands when comparing the forty trees per hectare to the hundred trees per hectare definition. Also as expected, the spread in the data increases with an increasing number of trees. Across all stands, the forty trees per hectare definition provided the largest mean dominant height value. When the definition is based upon the 85th percentile, the mean dominant height value fluctuates to either higher or lower than the 100 trees per hectare value. Across the study sites, tree density varies from 365 to 938 trees per hectare, allowing for the percentile method to be adjusted based on density changes, unlike the other two definitions. Because a percentile definition better reflects the relative differences across stands with large fluctuations in tree density, the 85th percentile definition will be used to investigate the impacts of soil and microtopography on dominant height and site index.

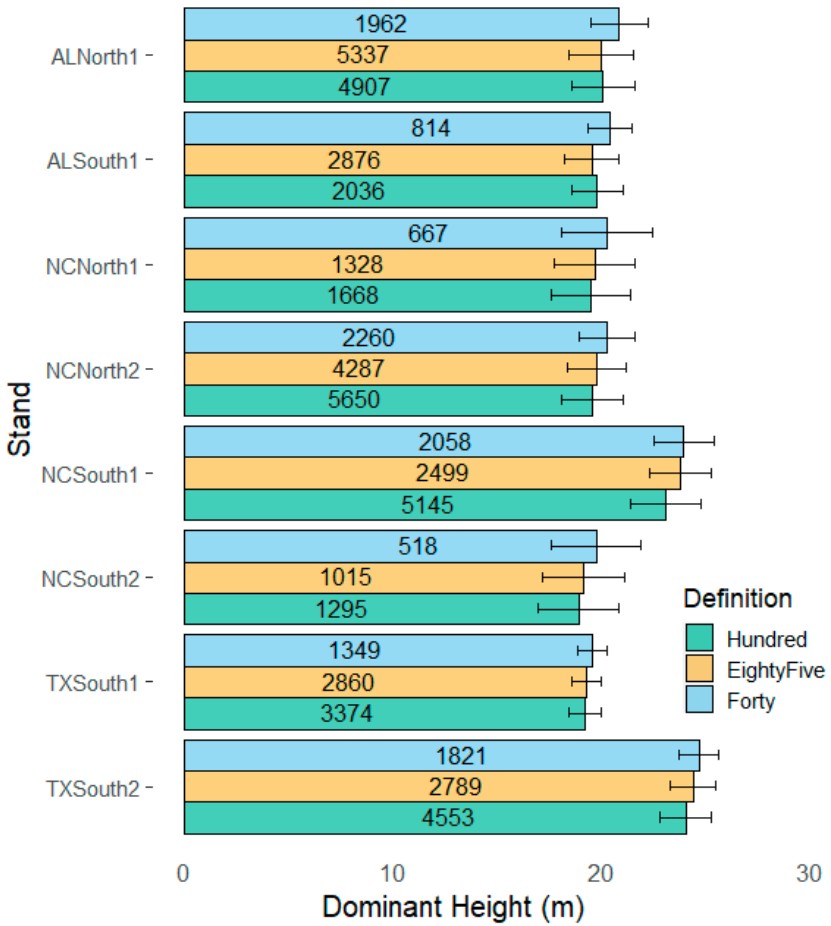

**Figure 3.** Dominant height values were computed at the stand scale using the following three definitions: (1) one hundred trees per hectare, (2) forty trees per hectare, and (3) the eighty-fifth percentile. The error bars represent the 95% confidence interval for the mean value from the simulated plot inventory data. The numbers listed inside the bars represent the number of trees used to calculate the dominant height per definition within a stand.

*3.2. Scale*

3.2.1. Soils

The number of soil types and their impact on dominant height varies by stand (Figure 4). The range of height for trees within the 85th percentile varied by state, with North Carolina having the widest range and Alabama having the narrowest range. In stand ALSouth1, two soil types were found to be statistically different from the full stand estimate for mean dominant height and two soil types were not: C2WextScPD and B1WeaoBgPD (*p*-values of 0.94 and 0.64). Soil type C2WextScPD, found in stand ALNorth1, was not significantly different from the full stand (*p*-value 0.94) but B4SyxfSrPD was (*p*-value < 0.01). Across the four stands within North Carolina, there was one soil type that was not statistically significant for one stand but was significant for another, B2WexvAuPD (*p*-values < 0.01 and 0.93). Stand NCNorth2 had only one soil type and was not included in the analysis. The boxplots show that two of the soil classes in North Carolina have greater variability when compared to the others, E2WykoAuPD and B2WexvAuPD. The greatest differences in soil types can be seen in the state of Texas. Three soil types were found to not be statistically significant when compared to the full stand in Texas: C3WeioLbGF (*p*-value of 0.08) in stand TXSouth1, and C6MyioLbGF and C3PyaoLbGF (*p*-values of 0.61 and 0.17) in stand TXSouth2. Soil type C3PyaoLbGF was statistically significant for TXSouth1, but not for another stand (*p*-value < 0.01).

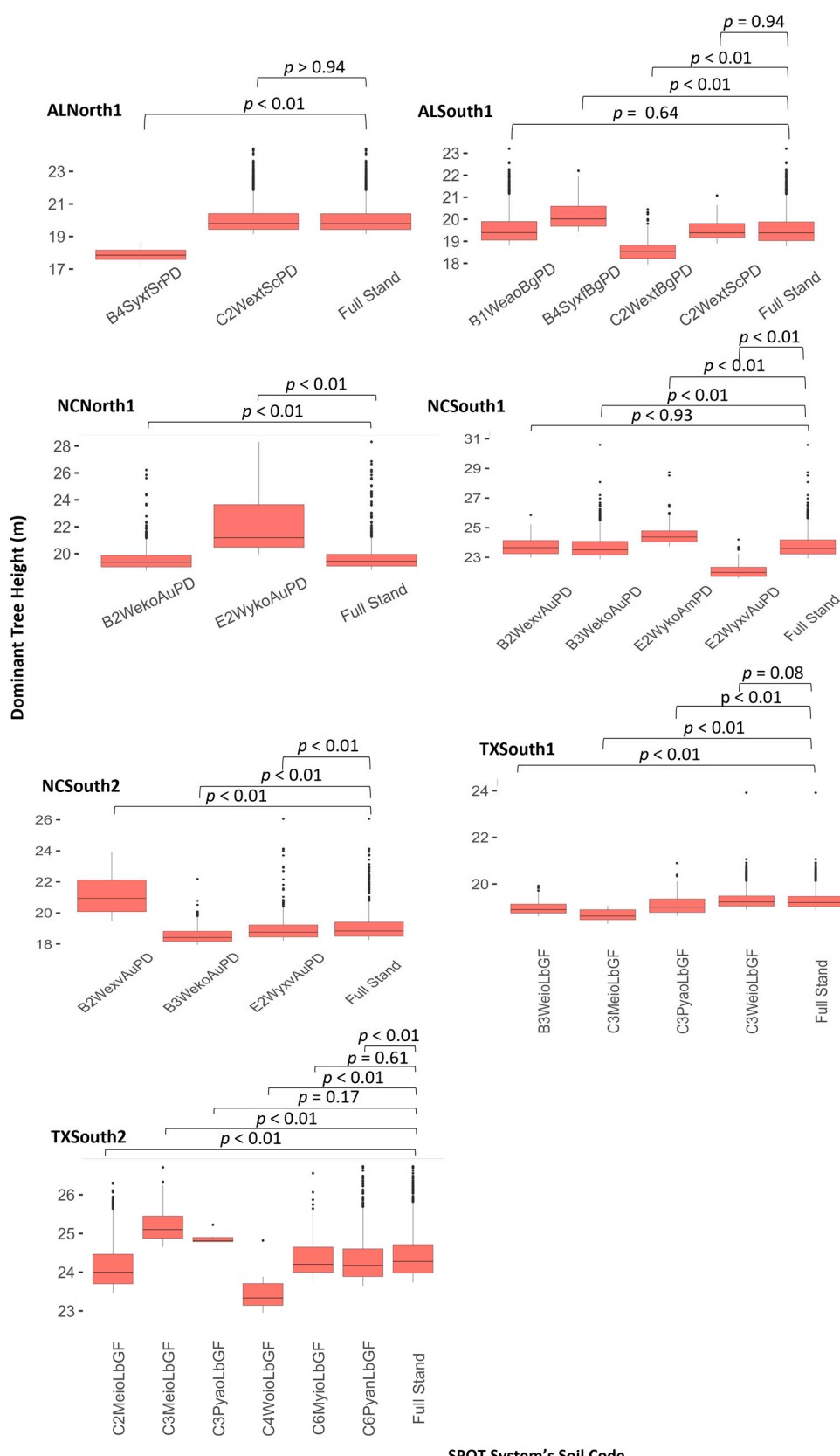

**Figure 4.** Average dominant height values for each soil type within a stand were compared. Any soil types with less than five (5) values were excluded from the graphic. Dunnett's test results were calculated using the means for each soil class compared to the full stand scale values for each stand.

### 3.2.2. Microtopography

The results for microtopography indicate that Alabama has the smallest range of height values for the 85th percentile trees within the microtopography scale (Figure 5). The *p*-values indicate that all microtopography class estimates for mean dominant height are statistically different from the full stand estimates in North Carolina. In stand ALSouth1, the mean dominant height for those trees found on a flat is not statistically different from the full stand estimate (*p*-value of 0.37), while those on mounds or in valleys show statistical differences in height. All microtopography class estimates are statistically different from the full stand estimates in stand ALNorth1. A different trend is seen in Texas, where the mean dominant height for those trees found in valleys and on a flat is not different from the full stand estimate in the stand TXSouth1 (*p*-values of 0.22 and 0.07). For stand TXSouth2, the mean dominant height for trees found on a flat is not statistically different from the full stand estimate (*p*-value of 0.39), while those on mounds or in valleys show statistical differences in height. In all states, the mean dominant height for trees in a valley is greater than the full stand estimate, with a very large difference found for Texas.

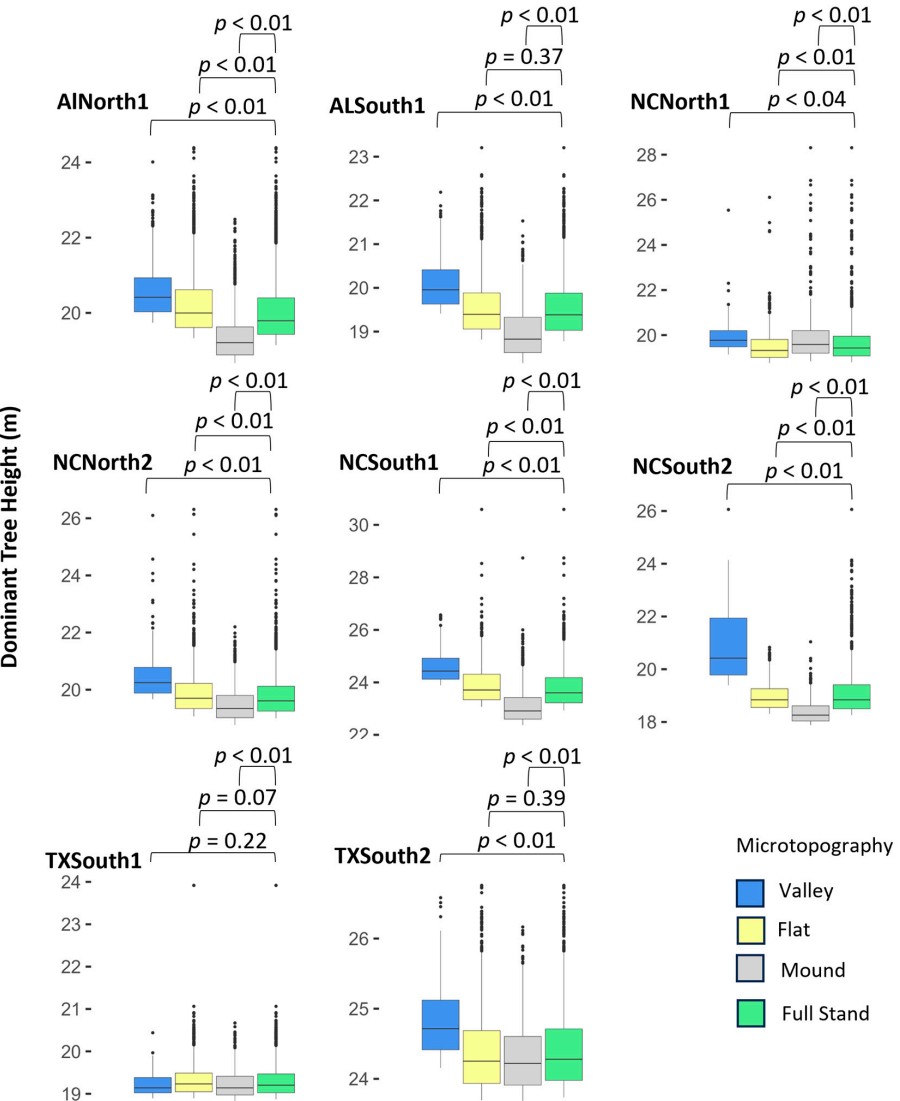

**Figure 5.** Average dominant height value for each microtopography class within a stand. The *p*-values provided at the top of the graph are for the null hypothesis that the mean differences are equal to zero. Dunnett's test results were conducted between the means for each soil class compared to the full stand scale values for each stand.

### 3.2.3. Soils and Microtopography

The heat maps below provide a visual representation of how soils, microtopography, and the interaction of soils and microtopography impact dominant height within the same region. For stand ALNorth1 in Alabama (Table 6), the percent change in average dominant height differs by 11 percent, while stand ALSouth1's percent change is 8 percent. Soil type C2WextScPD appears in both stands and shows no difference from the full stand estimate in either stand. Soil type B4SyxfSrPD shows the largest departure from the full stand estimate at −11%. Microtopography differences are consistent across both stands for estimates obtained for trees on a mound or in a valley, but the percent change for those trees in a flat area is different between the two stands. The mean dominant height for those trees in a valley is 3 percent greater than the full stand estimate, compared to the mean height of those in a mound, which is 3 percent less than the full stand estimate. The mean dominant height for those trees found on soil type B4SyxfBgPD in a valley is less than the full stand estimate, which is different from trees found in a valley on any other soil type. Dominant trees on a mound in the B4SyxfBgPD soil type are, on average, the shortest, and those in flat areas in the B4SyxfBgPD soil type are the tallest on average.

**Table 6.** Heat map for the stands in Alabama with each cell value representing the percent difference in average dominant height. The average dominant height for each scale was compared to the overall average for the full stand (labeled in grey). Green indicates a percent difference less than 3, red indicates a percent difference between 3 and 5 percent, and blue indicates a percent difference greater than 5 percent. Black indicates no trees in that combination of soil and microtopography.

| State | Stand | Soil | Flat | Mound | Valley | Sub Total |
|---|---|---|---|---|---|---|
| | | B4SyxfSrPD | | −9% | | −11% |
| | ALNorth1 | C2WextScPD | 1% | −3% | 3% | 0% |
| | | Sub Total | 3% | −3% | 3% | 20.2 m |
| AL | | B1WeaoBgPD | 0% | −3% | 3% | 0% |
| | | B4SyxfBgPD | 4% | −12% | −2% | 3% |
| | ALSouth1 | C2WextBgPD | −7% | −7% | | −5% |
| | | C2WextScPD | −1% | −8% | 2% | 0% |
| | | Sub Total | 0% | −3% | 3% | 19.6 m |

A greater range in percent difference is seen across the soil types from North Carolina as compared to Alabama. Two soil types show large positive percent differences: E2WykoAuPD (13 percent) and B2WexvAuPD in stand NCSouth2 (11 percent); however, B2WexvAuPD shows a slightly smaller percent difference in NCSouth1 (3 percent). Soil type E2WyxvAuPD shows a large negative percent difference at −7 percent. Trees located in a valley show a percent increase in average dominant height when compared to the full stand estimate; those in a mound have a negative percent difference; and those on a flat show little percent difference. Dominant trees on a mound in the E2WykoAuPD soil type show a large increase in the percent difference while across all other soil types, dominant trees located on a mound are shorter on average when compared to the full stand (Table 7).

The stands in Texas show the least amount of variability (Table 8). Percent differences are more pronounced across different soil types than microtopography. The stands in Texas have the smallest elevation range. Dominant trees located on flat surfaces within the C3MeioLbGF soil type are the shortest on average when compared to the full stand and those within the C3MeioLbGF soil type are the tallest.

### 3.3. Site Index

The Wilcoxon test indicates that when compared to the full stand estimate for site index, the weighted estimates using the soils and the interaction of soils and microtopography are statistically different for an alpha level of 0.05. In addition, the interaction of soils and microtopography is statistically different from all other scales but not when compared to the plot simulation values (Figure 6 and Table 9).

**Table 7.** Heat map for the stands in North Carolina with each cell value representing the percent difference in average dominant height. The average dominant height for each scale was compared to the overall average for the full stand (labeled in grey). Green indicates a percent difference less than 3, red indicates a percent difference between 3 and 5 percent, and blue indicates a percent difference greater than 5 percent. Black indicates no trees in that combination of soil and microtopography.

| State | Stand | Soil | Flat | Mound | Valley | Sub Total |
|-------|-------|------|------|-------|--------|-----------|
| NC | NCNorth1 | B2WekoAuPD | −1% | 0% | 1% | −1% |
| | | E2WykoAuPD | 8% | 16% | | 13% |
| | | Sub Total | −1% | −1% | 2% | 19.7 m |
| | NCNorth2 | B2WekoAuPD | 0% | −2% | 3% | 0% |
| | | Sub Total | 0% | −2% | 3% | 19.8 m |
| | NCSouth1 | B2WexvAuPD | 1% | −5% | −3% | 0% |
| | | B3WekoAuPD | 0% | −3% | 3% | 0% |
| | | E2WykoAmPD | 3% | 2% | 5% | 3% |
| | | E2WyxvAuPD | −7% | −7% | | −7% |
| | | Sub Total | 0% | −3% | 3% | 23.8 m |
| | NCSouth2 | B2WexvAuPD | 1% | −2% | 18% | 11% |
| | | B3WekoAuPD | −2% | −5% | 0% | −3% |
| | | E2WyxvAuPD | −1% | −4% | 6% | −1% |
| | | Sub Total | −1% | −4% | 10% | 19.1 m |

**Table 8.** Heat map for the stands in Texas with each cell value representing the percent difference in average dominant height. The average dominant height for each scale was compared to the overall average for the full stand (labeled in grey). Green indicates a percent difference less than 3, red indicates a percent difference between 3 and 5 percent, and blue indicates a percent difference greater than 5 percent. Black indicates no trees in that combination of soil and microtopography.

| State | Stand | Soil | Flat | Mound | Valley | Sub Total |
|-------|-------|------|------|-------|--------|-----------|
| TX | TXSouth1 | B3WeioLbGF | −1% | 1% | −3% | −2% |
| | | C3MeioLbGF | −6% | −2% | 0% | −3% |
| | | C3PyaoLbGF | −1% | 0% | 1% | −1% |
| | | C3WeioLbGF | 0% | 0% | −1% | 0% |
| | | Sub Total | 0% | 0% | 0% | 19.3 m |
| | TXSouth2 | C2MeioLbGF | −2% | −2% | 2% | −1% |
| | | C3MeioLbGF | 4% | 3% | 2% | 3% |
| | | C3PyaoLbGF | | | | 2% |
| | | C4WoioLbGF | −4% | −5% | | −4% |
| | | C6MyioLbGF | −1% | 0% | 2% | 0% |
| | | C6PyanLbGF | 0% | −1% | 0% | 0% |
| | | Sub Total | 0% | 0% | 2% | 24.4 m |

**Table 9.** Comparison of weighted site index estimations across all stands for each scale with the Wilcoxon test. *p*-values listed in bold indicating statistical significance.

| Stand Scale | Soils | Microtopography | Soils and Microtopography | Plot Simulation |
|-------------|-------|-----------------|---------------------------|-----------------|
| Full Stand | **0.0225** | 0.0547 | **0.0078** | 0.2500 |
| Soils | | 0.6406 | **0.0078** | 0.3828 |
| Microtopography | | | **0.0225** | 0.3125 |
| Soils and Microtopography | | | | 0.6406 |

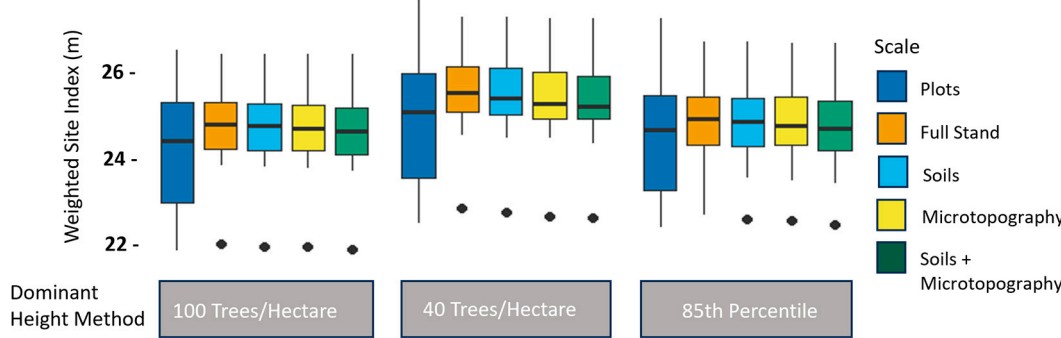

**Figure 6.** Weighted dominant height values computed using stand-scale level (i.e., full stand, soil classification, microtopography classification, and combined soil and microtopography classification) within three definitions (i.e., forty trees per hectare, the eighty-fifth percentile, and a hundred trees per hectare).

## 4. Discussion

### 4.1. Productivity Definition

Measures of site quality provide land managers with information that is critical to forestland decision-making processes. Improvements in the accuracy and resolution of site quality offer the potential to make better decisions regarding tree density choices, potential localized growth rates, and nutrient availability or response potential from additives such as fertilizer or herbicide treatments. An increased understanding of these conditions within a forest stand provides opportunities to minimize costs and increase profitability, as well as influence additional landowner objectives such as forest health and aesthetics, wildlife preservation, and/or hydrologic function. Profitability is often assessed as the amount of current or future merchantable volume available within a stand at a given point in time. Traditionally, site index has been used as a measure of site quality and has been defined and spatially confined to within stand boundaries within the southeastern U.S. Burkhart and Tomé [5] define a stand as a contiguous group of trees, sufficiently uniform in age–class distribution, species composition, and structure, growing on a site of sufficiently uniform quality to be a distinguishable unit. Stand boundaries in the southern U.S. typically reflect localized geographic features such as roads, land ownership, fire breaks, water bodies, and other physical land features, rather than uniform productivity. The use of remote sensing data such as satellite imagery, PhoDAR point clouds, and LiDAR point clouds creates a unique opportunity to assess information at a scale finer than the stand level. The use of LiDAR point clouds to create tree maps and assess microtopography moves the metrics available to forestland management decisions closer to those available in precision agriculture, thus providing the ability to assess conditions and apply management activities at finer resolutions. This creates the need to understand how traditional metrics such as dominant height and site index perform at scales other than the stand scale.

### 4.2. Dominant Height

Dominant height performance is linked directly to the definition used for the calculation. Sharma et al. [41] investigated seven definitions for dominant height across thinned and unthinned loblolly pine stands and determined that, in most cases, all seven definitions produced different results. This study indicates that for stands with sharp differences in density (e.g., trees per hectare), a more consistent result is achieved using a definition for dominant height that is based on a percentile of the trees than when using a dominance selection based on trees per hectare counts [42].

Traditionally, dominant height was determined by measuring dominant trees within a stand. In some cases, dominant-height trees were selected by placing field plots within the stand. Determining the correct plot size can be challenging, and bias can be introduced if the plot size does not appropriately capture the distribution of dominant height within

the stand [10,41]. This study found that the dominant height was shorter for the simulated plots when compared to using the full stand tree list developed from the LiDAR point cloud. This difference in dominant height estimate is likely due to selection bias as the grid of plots was spaced evenly throughout the stand, capturing a measure of within-stand height heterogeneity. García and Batho [43] define this selection effect as the phenomenon of when larger trees within the stand are not located on the plot.

*4.3. Influence of Site Variability on Dominant Height*

Tree height in the southern U.S. is often influenced by microtopography and soil variations, which are foundational to defining microsite conditions. Microtopography is defined as influential topographic variability for individual trees, and it has been described as small-scale (<2 m) surface variation within an elevation range [44]. Additionally, variations in soil type (texture, structure, etc.) are often present within traditionally defined stand boundaries. Between soil types, different soil properties affect vegetation. For example, soil depth has been found to be an indicator of effective rooting volume [45]. Similarly, Koirala et al. [46] found strong associations between tree height and water-holding capacity in surface soil conditions related to soil permeability and aeration in northeastern Texas. Studies from other geographic regions, such as those by Zhao, Y. et al. [47], have observed that relative bulk density can limit total tree height for Douglas-fir and lodge pole pine in British Columbia. The influences of soil type and condition suggest that a combination of shallow soil depth, low rainfall, and less fertile horizons may slow tree growth [48]. Thus, understanding the soil attributes across a landscape would lead to improved site index predictions [49]. Even in relatively homogenous sites, when other resources are limited, small variations in soil nutrients or water availability can positively influence microsite productivity [49].

In addition to soils, microtopographic conditions impact dominant height at a finer scale. Small variations in soil drainage within pine plantations located within the southeastern U.S, Western Gulf Coast, are not always caused by broad changes in slope but rather small topographic changes associated with randomly distributed "pimple mounds", circular to elliptical soil mounds ranging from 0.5 to 1.5 m in height and 10 to 30 m in diameter [50]. Additionally, the presence of a floodplain can lead to an accumulation of soil organic carbon (SOC) and an increased cation exchange capacity (CEC), providing vegetation with additional nutrient uptake abilities. This contrasts with higher elevation, ridge, and spur positions, where a lower CEC may exist due to rainwater leaching, erosion, and past land-use practices that further impact the availability of nutrients such as potassium (K), which is critical to tree growth [51,52]. Tree growth for some species has been shown to increase in relation to sediment deposition areas and the presence of more neutral pH levels [53], which can positively affect CEC. Additionally, during seasonal or extreme droughts, microtopographic conditions can influence water availability to trees in lowland areas, where alluvial or erosive soil deposition has increased soil depth and texture, thereby influencing the water-holding capacity of some areas and supporting the recovery and resilience of soil health [54]. These microtopographic changes and local topography influences influence tree growth, soil fertility, and water availability at multiple scales, even within individual inventory field plots [55,56]. Site index maps have been used to improve dominant tree height estimates for growth and yield modeling [49,57–60]. Therefore, the inclusion of high-resolution ALS data for tree height, tree density, and microtopography could help to predict soil features, like water availability and texture, since soil nutrition partly depends on the topographical position. The influence of soil and microtopography on tree height has been acknowledged throughout many forested regions, and thus the heterogeneity within a forester-drawn stand is also an aspect of forest management that needs to be considered. If dominant-height trees are not found evenly throughout a stand, there may be LiDAR-derived values that can be used to delineate stands to more biologically representative areas. Similarly, Liu and Burkhart [61] found that increments of total height were spatially autocorrelated.

Using ancillary data to map soil types and define dominant trees at a different scale can lead to better dominant height estimates. We found that soil type influenced dominant height estimates. For example, the C2WextScPD soil type in Alabama produced a higher dominant height than the B4SyxfSrPD soil type. The C2WextScPD soil type has a stronger nutrient availability of C, Fe, K, and Mg, a well-drained soil profile characterized by a coarse loamy texture, and a shallower depth due to the argillic horizon. In contrast, the B4SyxfSrPD soil type has good nutrient availability for K, drains poorly, and possesses a deeper depth due to the increase in clay. Kelting et al. [62] found that for young loblolly pine, site preparation treatments, and soil productivity indicators explain 87% of the variation in tree height; and Kelting et al. [63] found that surface soil depth had a weak positive effect on tree growth. At the study site in Texas, the lowest dominant height values were found in well-drained soil with a coarse loamy texture, while the soils with wetter drainage conditions had higher dominant height values. Pangle and Seiler [64] found that soil moisture limitations in coarse loamy soils can limit microbial activity in loblolly pine. These results were somewhat anticipated given the body of literature defining the influences of soil texture, structure, CEC, and water-holding capacity on vegetation development within both commercial forest and agricultural systems.

Incorporating microtopography metrics resulted in an improvement in the overall location accuracy of trees exhibiting a dominant height. We found that across all three states, dominant-height trees located in a valley or lowland positions showed greater total height-by-age than those found on a mound or in a flat area of similar age. These results are similar to those described by Borders et al. [52], who found that for loblolly pine sites located near each other in the Dixon State Forest near Waycross, Georgia, USA, slight elevation changes create distinct site characteristics. Lorio et al. [65] observed that for loblolly pine located on wet sites, variations in mycorrhizal surface area were dependent upon localized differences in microtopography (pimple mounds up to 2 m). The heat maps developed in this study highlight differences in combinations of soil type and microtopography impact in terms of measured total height. Changes in total height are observed within and between stands when either the soil type or the microtopography are held constant. Additionally, the weighted site index Wilcoxon tests indicate significant differences in weighted site index when the interaction of soil and microtopography are considered together from location to location. Care should be taken when considering estimates of tree dominance, and ultimately site index, at any scale to ensure that the area sampled for each class is large enough to provide a representative sample and considers the heterogeneity of soil type and microtopography throughout [66].

*4.4. Precision Forestry Using Remote Sensing Data*

Conventionally, forest management has focused on using sampled, mean, and stand-level estimates of forest attributes for planning and management activities within growth and yield models of projected performance. The research here suggests LiDAR data can enable a total stand population to be assessed and identify differences in valuable metrics such as dominant tree height and relative spatial positioning at scales finer than the stand-level. This differentiation enables us to consider tailored silviculture planning and management practices such as targeted fertilizer applications to improve forestland resource investments. Incorporating ancillary data such as soil type maps and microtopography can improve estimates of variables such as dominant height and site index, which are often important drivers within growth and yield models. Future research should consider the potential of using measured tree growth from LiDAR populations to improve soil attribute mapping for aspects such as depth and water-holding capacity. From the results of this research, it is suggested that, within the southeastern U.S., stands of loblolly can potentially be managed at different scales to optimize resource allocation in scenarios of increasing fertilizer prices and uncertainty in timber prices [4].

## 5. Conclusions

ALS LiDAR data collected at or above the densities presented in this work offer a robust opportunity to characterize total forest stand populations and provide information at finer scales than previously available to inform forest management practices. The interaction of soil conditions and microtopography can play a meaningful role in the average height of dominant trees within loblolly stands similar to those in this study. However, the results of this work indicate that slope position and tree density also play key roles in the height and growth within plantations similar to these. Forestland managers should consider soil and microtopography within management decisions because, when included in a weighted site index measure, the resulting values were statistically different from standard practices within these sites. Additionally, it may be beneficial to employ stratification techniques when measuring site quality and productivity to prevent bias in measurements of dominant height when employing them for determining site index, especially in stands with a wide variety of soil types and microtopographies [67]. Publicly available LiDAR data, regardless of their low pulse densities, can provide valuable microtopography information (<2 m DTM), depending on the composition and density of the understory vegetation. However, LiDAR data with an increased pulse energy and greater pulse densities has been shown to capture more accurate tree heights and improved tree counts compared to publicly available datasets. The cost of these mostly private collections may, however, be prohibitive for some project scales and restrict the ability to create high-quality individual tree segmentation maps. Lastly, while access to repeated LiDAR datasets may be limited in many geographic regions, opportunities for using PhoDAR or surface models of tree canopies in conjunction with LiDAR datasets may provide an option for assessing tree height changes over time at a stand or sub-stand resolution.

**Author Contributions:** Conceptualization, I.R.-G., L.C.R. and R.L.C.; methodology, I.R.-G., L.C.R. and R.L.C.; software, I.R.-G., L.C.R. and M.J.S.; validation, I.R.-G. and L.C.R.; formal analysis, I.R.-G. and L.C.R.; investigation, I.R.-G. and L.C.R.; resources, L.C.R., R.L.C. and J.S.B.; data curation, I.R.-G.; writing—original draft preparation, I.R.-G.; writing—review and editing, L.C.R., R.L.C., M.V.C. and J.S.B.; visualization, I.R.-G.; supervision, L.C.R., R.L.C. and J.S.B.; project administration, R.L.C.; funding acquisition, R.L.C. and J.S.B. All authors have read and agreed to the published version of the manuscript.

**Funding:** Funding for this work was provided in part by the McIntire-Stennis Program of the National Institute of Food and Agriculture, U.S. Department of Agriculture Accession No. 1025518, and the National Science Foundation Center for Advanced Forestry Systems Award 1916552. The use of trade names in this paper does not imply endorsement by the associated agencies of the products named nor criticism of similar ones not mentioned.

**Data Availability Statement:** The raw data supporting the conclusions of this article will be made available by the authors upon request.

**Acknowledgments:** We appreciate the support from the Forest Productivity Cooperative. We gratefully acknowledge the support provided by the Department of Forestry and Environmental Resources at North Carolina State University, Virginia Department of Forestry, the Department of Forest Resources and Environmental Conservation at Virginia Polytechnic Institute an State University, the Departamento de Silvicultura, Facultad de Ciencias Forestales, Universidad de Concepcion, and the Federal University of Lavras.

**Conflicts of Interest:** The authors declare the following financial interests/personal relationships which may be considered as potential competing interests: Rachel L. Cook reports financial support was provided by National Science Foundation. Ivan Raigosa-Garcia reports financial support was provided by National Institute of Food and Agriculture. The additional authors declare that they have no known competing financial interests or personal relationships that could have appeared to influence the work reported in this paper.

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
