# Peer review of "Rethinking Productivity Evaluation in Precision Forestry through Dominant Height and Site Index Measurements Using Aerial Laser Scanning LiDAR Data"

_forests, doi:10.3390/f15061002_

Round 1
Reviewer 1 Report
Comments and Suggestions for Authors
This manuscript aims to assess the effect of scale on dominance height and site index measures within loblolly pine plantations using LiDAR and ancillary data, falls within the scope of the journal's research, and has some application value considering the practical needs of forestry management. But this manuscript has the following problems:
1. It is recommended to concisely summarize the abstract with a clear statement of the significance of the study and the current challenges.
2. The manuscript is of poor readability. The introduction section is not logical enough and the introduction of each section is scattered. The methods section does not point out the shortcomings of the existing methods and the methods to be used in this study.
3. Regarding the extraction accuracy of DTM, the manuscript should state the accuracy evaluation method and results
4. Please add the criteria for selecting the ITC segmentation method, and whether the accuracy is reliable after segmentation are not reflected in the manuscript.
5. The discussion section looks like the Introduction, why are the conclusions of previous researchers not presented in the Introduction section?
6. The authors should add recommendations for practical management based on the conclusions drawn from this study.
7. the authors should improve the detailed analysis process to conclude this paper.
8. the authors should have a separate section to fully summarize the experimental findings and to write about the innovations of this study.
9. the authors should be more careful with the formatting of the article, which contains several formatting errors. For example, in section 2.1, all the references failed to be inserted and the formatting of the tables is different from the rest of the paper. Section 2.2 lacks numbering. References 35 are not formatted consistently. Multiple reference insertion failures.
Based on the above comments, I believe that the innovative nature of this study is not sufficient, but the experimental data processing is adequate. It is recommended for major revision.
Comments on the Quality of English Language
good
Author Response
First, thank you for taking the time to review this manuscript.
Comment 1:
Thank you for this suggestion. The abstract was edited to highlight the significance of the study and the current challenges, stating that: “These results highlight how precision forestry may provide a unique opportunity for assessing soil and microtopographic information to optimize resource allocation and forest management at an individual tree scale in a scarce higher-priced fertilizer scenario”. The last sentence added to the abstract highlights the importance of the study in a scenario of increased fertilizer prices. Where resource optimization must be a priority sub stand characterization is vital to reach that goal.
Comment 2:
Thank you for your suggestions. The manuscript's readability has been improved. The introduction section has been rearranged to be more logical, and we added subheadings to help improve the flow. The subheading starts with Loblolly pine's importance for the US economy and then transitions to dominant height as a measure of productivity. Once the unit of measurement is described in this paragraph, we proceed to introduce stand characterization with remote sensing data section and finally, we introduce the section on the use of remote sensing in precision Forestry.
Comment 3:
We did not have ground control points to estimate the accuracy of the DTM by comparing lidar-derived DTM to a control DTM. However, all airborne acquisitions used on-board aircraft navigation and real-time kinematic geographic positioning system with a precision of ≤10 cm and positional accuracy of ≤5 cm using RiACQUIRE (version 1.3.7. RIEGL, 2022). To highlight this comment, we added the following paragraph: “All airborne acquisitions used on-board aircraft navigation and real-time kinematic geographic positioning system with a precision of ≤10 cm and positional accuracy of ≤ 5 cm using RiACQUIRE [28]”
Comment 4:
Thank you for this question. We selected the ITC method because it was specifically developed in loblolly pine and it has a high detection accury in low density stands (618 trees/ha). To Clarify this in the manuscript we added the following paragraph to the section 2.3: “ITC segmentation method created by Sumnall et al. [15] for loblolly pine. The detection of stem locations in this method relies on exploiting the high pulse density of the LiDAR da-ta. ITCS delineation is based on segmenting the canopy height model (CHM) [12] using previously detected stems as initial markers. The author reported an average proportion of missed alive trees of around 8%; however, in low tree densities (618 trees/ ha), the method only missed 1-3%.”
Comment 5:
Thank you for this suggestion, we have moved text between the introduction and discussion sections in order to provide additional clarity and enhance the flow of the paper. In addition, we have generally reorganized these sections with additional subtitles to improve the organization and flow of the manuscript.
Paragraph moved from Intro to discussion: “Tree height in the southern U.S. is often influenced by microtopography and soil variations, which are foundational to defining microsite conditions. Microtopography is defined as influential topographic variability for individual trees, and it has been de-scribed as small-scale (<2 m) surface variation within an elevation range [45]. Additional-ly, variations in soil type (i.e., texture, structure, etc.) are often present within traditionally defined stand boundaries. Between soil types different soil properties affect vegetation. For example, soil depth has been found to be an indicator of effective rooting volume [46]. Sim-ilarly, Koirala et al. [47] found strong associations between tree height and water-holding capacity in surface soil conditions related to soil permeability and aeration in northeast-ern Texas. Studies from other geographic regions, Zhao, Y. et al. [48] have observed that relative bulk density can limit total tree height for Douglas-fir and lodge pole pine in Brit-ish Columbia. Understanding the influences of soil type and condition suggests a combi-nation of shallow soil depth, low rainfall, and less fertile horizons may slow tree growth [49]. Thus, understanding soil attributes across a landscape would lead to improved pre-dictions of site index [50]. Even in relatively homogenous sites, when other resources are limited, small variations in soil nutrients or water availability can positively influence microsite productivity [50].
In addition to soils, microtopographic conditions impact dominant height at a finer scale. Small variations in soil drainage within pine plantations located within the south-eastern U.S, Western Gulf Coast, are not always caused by broad changes in slope but ra-ther small topographic changes associated with randomly distributed “pimple mounds”, circular to elliptical soil mounds, ranging from 0.5 to 1.5 meters in height and 10 to 30 meters in diameter [51]. Additionally, the presence of a floodplain can offer an accumula-tion of soil organic carbon (SOC) and an increased cation exchange capacity (CEC) providing vegetation with additional nutrient uptake abilities. This contrasts with higher elevation, ridge and spur positions, where lower CEC may exist due to rainwater leaching, erosion, and past land-use practices that further impact the availability of nutrients such as potassium (K) which is critical to tree growth [52,53]. Tree growth for some species has been shown to increase in relation to sediment deposition areas and the presence of more neutral pH levels [54], which can positively affect CEC. Additionally, during seasonal or extreme droughts, microtopographic conditions can influence water availability to trees in low-land areas, where alluvial or erosive soil deposition has increased soil depth and texture thereby influencing the water-holding capacity of some areas and supporting the recovery and resilience of soil health [55]. These microtopographic changes and local to-pography influences influence tree growth, soil fertility, and water availability at multiple scales, even within individual inventory field plots [56,57]. Site index maps have been used to improve dominant tree height estimates for growth and yield modeling [50,58-61]. Therefore, the inclusion of high-resolution ALS data for tree height, tree density, and mi-crotopography could help to predict soil features, like water availability and texture, since soil nutrition partly depends on the topographical position. The influence of soil and mi-crotopography on tree height has been acknowledged throughout many forested regions and thus the heterogeneity within a forester-drawn stand is also an aspect of forest man-agement that needs to be considered”
Comment 6:
Thank you for this suggestion. We have revised the conclusions section accordingly and improved the recommendations for practical management based on the conclusions of the study. We recommend practical management based on the conclusions, the use of Geological and soil maps in addition to publicly available LiDAR help to improve productivity estimation and allocate resources in a more efficient manner. We have added the following text to the conclusions section:
“Forestland managers should consider soil and microtopography within management decisions because when included in a weighted site index measure the resulting values were statistically different from standard practices within these sites. Additionally, it may be beneficial to employ stratification techniques when measuring site quality and productivity to prevent bias in measurements of dominant height when employing them for determining site index, especially in stands with a wide variety of soil types and microtopography [65][60]. Publicly available LiDAR data, regardless of its low pulse densities, can provide valuable microtopography information (<2m DTM), depending on the composition and density of the understory vegetation. However, LiDAR data with increased pulse energy and greater pulse densities has been shown to capture more accurate tree heights and improved tree counts over publicly available datasets. The cost of these, mostly private collections may however be cost-prohibitive for some project scales and restrict the ability to create high-quality individual tree segmentation maps. Lastly, while access to repeated LiDAR datasets may be limited in many geographic regions, opportunities for using PhoDAR or surface models of tree canopies in conjunction with LiDAR datasets may provide an option for assessing tree height changes over time at a stand or sub-stand resolution”.
Comment 7:
Thank you for this suggestion. We have revised the analysis section. We added the description of the analysis for the site index in section 2.4.3 which was missing from the methods: “ The Wilcoxon test [41] was conducted to determine if there was a difference among weighted site index estimations across all stands for each scale. For each test, an al-pha-level of 0.05 was used to determine statistical significance.”
Comment 8:
We have bolstered the introduction to highlight the specific contributions of this analysis in the following paragraph: We added the section to highlight the findings of the study: “This study makes several contributions to literature. First, we provide an application that uses ALS LiDAR and topographic data to characterize within-stand productivity for loblolly pine plantations – a globally important supply source of timber. Second, our analysis demonstrates that while dominant height definitions may be consistent, the per-centile definition does a better job representing dominant height in contrasting tree densi-ties. The study also reveals that soil and topography play an important role in dominant height, even at the sub-stand level, and it should not be overlooked. We also develop and demonstrate a method to stratify and weight site productivity, and it provides tools to in-corporate soils and microtopography in management decisions. Finally, we discuss the implications of our findings for forest managers interested in precision silviculture.”
Comment 9:
Thank you for your comment. We have revised the formatting of the document to correct formatting errors. The references in section 2.1 have been fixed and are now correctly formatted. Section 2.2 has been revised, and it now includes numbering. The references have been reviewed and fixed in the sections where the insertion failed or formatting was not consistent. Some errors seem to be related to the presence of hyperlinks within the file (Error! Reference source not found.), however, these issues have also been addressed and highlighted in the manuscript to ensure they are fixed.
Reviewer 2 Report
Comments and Suggestions for Authors
Find my comments in the pdf file. I wish you good luck.
1
Comments
Estimation of dominant tree height, site index as well as other attributes in plantations is a relatively easy thing to do. Have you considered to use actual mixed managed forest plots do examine that? In this case you could also use aerial photogrammetry to do so. Please explain what led you to follow such a methodological approach?
Line 25: Mention what kind of LiDAR source did you use (terrestrial or aerial or both). You may also need to clarify that in your title as well.
Lines 57-59: This sentence needs a reference.
In the fourth paragraph you mentioned what ALS can offer in terms of extraction metrics accuracy but in my opinion try to create separate paragraphs to holistically approach this topic, by mentioning what TLS
and ALS (LIDAR) can offer and how fusion of those two approaches can solve such issues (related to forestry structure).
Line 181 (soils and geology): How recent those soil maps are? Provide some date and data information.
In subsection 2.3. Provide some additional information regarding the flights. Who and how they have been conducted (manually, semi-automatically, at what altitude, flight pattern, etc.).
What was the reason for using different aerial sensors, what is that you explicitly wanted to achieve and why in different years (table 4)? That could be also related to my first comment where you could also use latest multispectral cameras to see how they perform especially with the low density lidar sensors that you have used…
Why did you use R (for point cloud processing)? R is not the most optimized choice for handling large-scale lidar data and let`s say complex point cloud processing tasks. If you wanted to use
open source it should be done in either C language which is more advanced or python as a simpler solution because it contains better and more advanced libraries for this purpose. R is recommended for simple task like visualization but not segmentation and other processing purposes. Not to mention scaling and format compatibility issues. Please explain.
Maybe in your objectives (introduction) you can better clarify that you use multiple ALS sensors to assess the performance of the used sensors in terms of accuracy of tree height estimation.
I'd be happy to review the updated manuscript after it has been edited
Author Response
Comment: Estimation of dominant tree height, site index as well as other attributes in plantations is a relatively easy thing to do. Have you considered to use actual mixed managed forest plots do examine that? In this case you could also use aerial photogrammetry to do so. Please explain what led you to follow such a methodological approach?
Reply:
Thank you for this comment. We chose to use only LiDAR data because we aimed to examine what would happen if we had known all the trees in the stand, which can be possible using ALS LiDAR. Field estimates are resource intensive and would have taken too long given the number of trees, and we are focused on the ability of LiDAR to support precision forestry in managed plantation systems. We did not use photogrammetry because, with ground classification, we can get precise tree heights. Photogrammetry does not allow us to get tree height unless we have an initial DTM and control points, however, it may be useful for other studies to measure annual tree height growth. We had LiDAR clouds available from partner stakeholders that were not necessarily collected for this project in particular.
Comment: Line 25: Mention what kind of LiDAR source did you use (terrestrial or aerial or both). You may also need to clarify that in your title as well.
Reply:
We used ALS LiDAR. We clarified that in this new version
Comment: Lines 57-59: This sentence needs a reference.
Reply:
We added a reference for that sentence
Comment: In the fourth paragraph you mentioned what ALS can offer in terms of extraction metrics accuracy but in my opinion try to create separate paragraphs to holistically approach this topic, by mentioning what TLS
and ALS (LIDAR) can offer and how fusion of those two approaches can solve such issues (related to forestry structure).
Reply: Thank you for this suggestion. We have added the following paragraph: "In most situations, estimating DBH is still a challenge from ALS data as data density plays a significant role in model accuracy and tends to decrease with closer proximity to the ground [12]. To solve this point density issue, terrestrial laser scanning (TLS) has been used to estimate DBH and individual tree volume [13,20-22]; however, scaling these methods from a few hectares to full stands or landscape scale has been challenging [20]. Nevertheless, the use of ultra-high-density drone LiDAR can produce clouds with thousands of points per square meter with low altitude flights and wide scan angles that resolve individual stem and branch structures [20]"
Comment: Line 181 (soils and geology): How recent those soil maps are? Provide some date and data information
Reply:
We added a reference for the age of the maps. We clarified that the soil maps, specifically the SSURGO database, represent more than 100 years of work.
Comment: In subsection 2.3. Provide some additional information regarding the flights. Who and how they have been conducted (manually, semi-automatically, at what altitude, flight pattern, etc.).
Reply:
We have provided additional information regarding the flights in section 2.3.
Comment: What was the reason for using different aerial sensors, what is that you explicitly wanted to achieve and why in different years (table 4)? That could be also related to my first comment where you could also use latest multispectral cameras to see how they perform especially with the low density lidar sensors that you have used…
Reply:
Thank you for these clarifying questions. The reason behind using different lidar sensors is that these data clouds are provided by a partnership between the Forest Productivity Cooperative (FPC) at NC State University and its stakeholders. These companies are flying independently as part of their own development and research program and provide the LiDAR clouds to the FPC for use in research applications. We did not use multispectral, TLS or super-high-density clouds because in this paper we were only interested in dominant height and site index as measures of productivity and not volume or basal area. We were investigating dominant height which can be acquired using mid-density ALS LiDAR data.
Comment: Why did you use R (for point cloud processing)? R is not the most optimized choice for handling large-scale lidar data and let`s say complex point cloud processing tasks. If you wanted to use
open source it should be done in either C language which is more advanced or python as a simpler solution because it contains better and more advanced libraries for this purpose. R is recommended for simple task like visualization but not segmentation and other processing purposes. Not to mention scaling and format compatibility issues. Please explain.
Reply:
Thank you for this recommendation. We found that using R was sufficient for the purposes of processing the LiDAR clouds used in this study, but we may investigate other methods for future applications to compare relative efficiency. The LAScatalog processing engine function allows the handling of large datasets that cannot be loaded all at once into memory. The LAScatalog engine splits the large datasets into tiles of desired sizes and has a buffer to prevent the absence of spatial context and rough edges, allowing the application of a workflow with automatic management of tile buffering. Additionally, there is a growing scientific literature that processes large datasets using the LiDR package in R.
Comment: Maybe in your objectives (introduction) you can better clarify that you use multiple ALS sensors to assess the performance of the used sensors in terms of accuracy of tree height estimation.
Reply:
Thank you for this suggestion. We have added additional clarification on this approach to the introduction and methods section, including the statement in subsection 2.3 that clarifies that our intent is not to compare data from different sensors but to take advantage of similar LiDAR data at different field sites provided by collaborators. “The choice of sensors was dictated by data availability rather than a deliberate intent to compare various sensors simultaneously.”